

# A semi-permeable attention network for ESG score prediction

Changlong Wang[1], Shixin Yang[2] and Yi Zhang[3]

[1] College of Business Administration, Wonkwang University, Jeonbuk, Republic of South Korea
[2] College of Business Administration, Shandong Polytechnic College, Shandong, China
[3] College of Accountancy, Shanghai University of International Business and Economics, Shanghai, China

## ABSTRACT

Environmental, Social, and Governance (ESG) metrics have become critical indicators of corporate sustainability, ethical behavior, and long-term financial performance. However, accurately predicting ESG scores remains challenging due to the tabular nature of ESG datasets and their small size, which often limits the effectiveness of traditional deep learning approaches. In this study, we propose an attention-based deep learning model specifically designed for tabular ESG data. Our model leverages the semi-permeable attention mechanism in the ExcelFormer architecture to selectively regulate feature interactions based on their predictive importance. This design enables the model to mitigate noise from less informative features while preserving critical dependencies within structured data. We evaluate our method on a simulated dataset comprising ESG and financial performance data from 1,000 companies across multiple industries and regions. The proposed model consistently outperforms traditional machine learning models and state-of-the-art tabular deep learning models, achieving the lowest errors across all ESG dimensions. Specifically, it attains a coefficient of determination of 0.7373 for overall ESG prediction, with a mean squared error of 0.0063. These results demonstrate the potential of attention-augmented tabular neural networks in advancing ESG forecasting, offering meaningful contributions to the field of sustainable finance.

Corresponding author
Yi Zhang, zhangluyi1020@163.com

# INTRODUCTION

Environmental, Social, and Governance (ESG) metrics provide frameworks for assessing corporations' sustainability and social responsibility across three key dimensions (*Tamimi & Sebastianelli, 2017*). Environmental metrics evaluate a company's ecological impact, with a focus on reducing carbon emissions and enhancing sustainability, which can improve reputation and attract investors (*Deng & Cheng, 2019*; *Zhou & Niu, 2024*). Social metrics consider stakeholder relationships, highlighting the importance of social equity and community engagement, which can enhance employee morale and productivity (*Przychodzen et al., 2016*; *Setiarini et al., 2023*). Governance metrics examine corporate processes, including board diversity and transparency, essential for effective

risk management and investor confidence (*Dziadkowiec & Daszynska-Zygadlo, 2021*). Integrating ESG factors into corporate valuation is increasingly recognized as a transformative approach to financial performance, indicating that strong ESG practices can mitigate investment risks and foster long-term value creation without sacrificing profitability (*Gao et al., 2023*; *Wang, 2024*). Overall, ESG metrics are vital for understanding corporate impact in today's economy and will continue to influence investment decisions and policy development as sustainability takes center stage (*Li et al., 2018*; *Jain, Sharma & Srivastava, 2019*).

Traditional methods for estimating and measuring ESG metrics encompass a variety of approaches and tools that shed light on a corporation's sustainability and ethical impact. While newer frameworks and methodologies have emerged, these traditional methods remain significant in the ESG landscape due to their established credibility and widespread use. One key method involves the utilization of ESG rating agencies, which play a prominent role in evaluating companies based on established criteria across environmental, social, and governance dimensions (*García et al., 2020*; *Mandas et al., 2023*). Organizations such as MSCI and Sustainalytics apply these established methodologies to assign ESG scores, allowing investors and stakeholders to assess corporate performance effectively. These scores are typically derived from a combination of annual sustainability reports, corporate filings, and third-party data sources; however, concerns about potential manipulation and comparability persist in the ESG discourse (*García et al., 2020*; *Mandas et al., 2023*; *Zenkina, 2023*). In addition to ESG rating agencies, corporate sustainability reporting is also used. This approach requires companies to voluntarily disclose their ESG practices and these reports often encompass metrics related to carbon emissions, labor conditions, governance structures, and community engagement. To aid in this effort, frameworks such as those provided by the Global Reporting Initiative (GRI) and the Sustainability Accounting Standards Board (SASB) guide companies on how to structure their disclosures. Consequently, these frameworks enhance stakeholder understanding of a company's long-term sustainability efforts (*Halid et al., 2023*; *Esposito et al., 2025*). Nevertheless, criticisms regarding the lack of standardization and the risk of greenwashing exist, where firms may present themselves in a misleading manner to appear more responsible than they truly are (*Esposito et al., 2025*). Moreover, Comparative Metrics and Indices, another method, evaluate ESG performance against industry benchmarks. By comparing a company's metrics to those of its peers, investors gain valuable insights into competitive advantages or disadvantages within the sector. This comparative analysis utilizes key performance indicators (KPIs) related to emissions reductions, employee diversity, and governance, enabling investors to assess performance on a relative scale (*Haryanto, 2024*). Furthermore, researchers have historically explored the financial performance correlation, such as Return on Assets (ROA) or market-based performance measures, as an effective indicators (*Xie et al., 2018*; *Garcia & Orsato, 2020*). Although some studies suggest statistically significant relationships, the evidence varies, revealing a complex interplay between sustainability practices and financial performance. Some research even indicates a negative correlation, highlighting that the relationship between ESG factors and financial health is multifaceted

and can shape investor perceptions (*Garcia & Orsato, 2020*). Finally, qualitative assessments are employed as another vital traditional method, where analysts conduct interviews, case studies, or thematic analyses to evaluate the broader cultural and ethical context of a company's operations. These methods can yield deeper insights, particularly regarding the social implications of corporate behaviors (*Monteiro, Moita Neto & da Silva, 2021*; *Wu, Zhai & Lv, 2024*).

Machine learning (ML) and deep learning (DL) have recently been applied in predicting ESG metrics, with an increasing focus on ethical and sustainable investing principles (*Minkkinen, Niukkanen & Mäntymäki, 2022*). These approaches offer significant technical advantages over traditional methods, particularly through DL models that excel at processing the complex, non-linear relationships inherent in ESG data (*Zhao, 2024*). The superior predictive accuracy of these models compared to conventional regression techniques stems from their ability to capture intricate patterns and interactions that traditional methods often miss (*Sætra, 2022*; *Xu, 2022*). The integration of ML and DL in ESG analysis provides substantial benefits for investment decision-making, including enhanced analytical rigor in ESG evaluations, better identification of risks and opportunities, and more systematic portfolio construction that effectively balances financial performance with ethical considerations (*Sætra, 2021*). These advanced approaches allow investors to gain deeper insights into environmental impacts, social responsibilities, and governance structures of their investment targets, while also improving compliance with sustainability standards and enhancing risk assessment accuracy (*Xu, 2022*). Moreover, ML and DL are also integrated into modern pipelines to accelerate ESG analysis processes, utilizing various techniques such as random forest feature selection and multi-layer perceptron models (*Capelli, Ielasi & Russo, 2021*). The versatility of these frameworks is demonstrated through cross-pollination of techniques from other fields, making them increasingly relevant for developing robust ESG predictive models (*Wang et al., 2023*; *Adeoye et al., 2024*). Overall, the integration of artificial intelligence (AI) advances in ESG prediction not only boosts analytical capabilities but also facilitates sustainable investment and corporate governance, though research continues to evolve in this rapidly developing field.

While ML and DL frameworks show promise for enhancing ESG prediction, they face several significant limitations that must be addressed. Data quality and availability present major challenges, as ESG data often lacks standardization across providers and organizations struggle to effectively integrate diverse data types such as news articles, social media posts, and satellite imagery that are crucial for robust analysis (*Juthi et al., 2024*; *Li et al., 2024*). Furthermore, the explainability of most ESG models remains a critical concern, as these models provide limited insight into how specific ESG factors influence predictions, which creates transparency challenges for stakeholders seeking to understand and trust these systems (*van der Heever et al., 2024*; *Giudici & Wu, 2025*). Additionally, the field suffers from imbalanced task prioritization, with frameworks often focusing heavily on environmental issues like pollution and carbon emissions while inadequately addressing social factors such as labor practices or community impact, potentially leading to skewed ESG assessments (*Li et al., 2024*). Despite these challenges, the growing

**Table 1 Numbers of samples and distribution of data used for modeling.**

| Score | Data | Number of samples |
|---|---|---|
| Environment | Training | 8,190 |
| | Validation | 990 |
| | Test | 1,100 |
| | Total | 10,280 |
| Social | Training | 8,190 |
| | Validation | 990 |
| | Test | 1,100 |
| | Total | 10,280 |
| Governance | Training | 8,190 |
| | Validation | 990 |
| | Test | 1,100 |
| | Total | 10,280 |

importance of computational advances in ESG analysis continues to motivate researchers to develop improved algorithms and methodologies.

In our study, we develop a model to predict ESG metrics based on ExcelFormer (*Chen et al., 2023*), a neural network architecture specifically designed for tabular data. ExcelFormer has been demonstrated to work effectively across all dataset sizes and types, from small to large tabular datasets where traditional DL models typically underperform. Unlike existing approaches that require intensive hyperparameter tuning, models based on this architecture can deliver competitive performance with pre-fixed parameters, providing a reliable and time-efficient solution that eliminates complex model selection for non-expert users while maintaining consistent results across diverse experimental conditions. Our model is also compared to traditional ML and other DL models for tabular data for fair performance assessment.

## DATASET

The study utilized a comprehensive simulated dataset encompassing the financial and ESG performance of 1,000 global companies spanning nine distinct industries and seven geographical regions over an 11-year period from 2015 to 2025. The dataset was collected from *Jiang (2024)*. This dataset incorporates realistic financial metrics including revenue, profit margins, market capitalization, and other key performance indicators, alongside comprehensive ESG indicators such as carbon emissions, resource utilization patterns, and detailed ESG scoring metrics. The dataset provides data for analyzing the relationship between environmental, social, and governance factors and corporate financial performance across diverse industrial sectors and regional markets. Comprehensive data preprocessing procedures were implemented to ensure data consistency and standardization, with categorical variables systematically encoded into numerical formats to optimize model compatibility and analytical performance. For each prediction task, data are split into a training, a validation, and a test with a ratio of (Table 1).

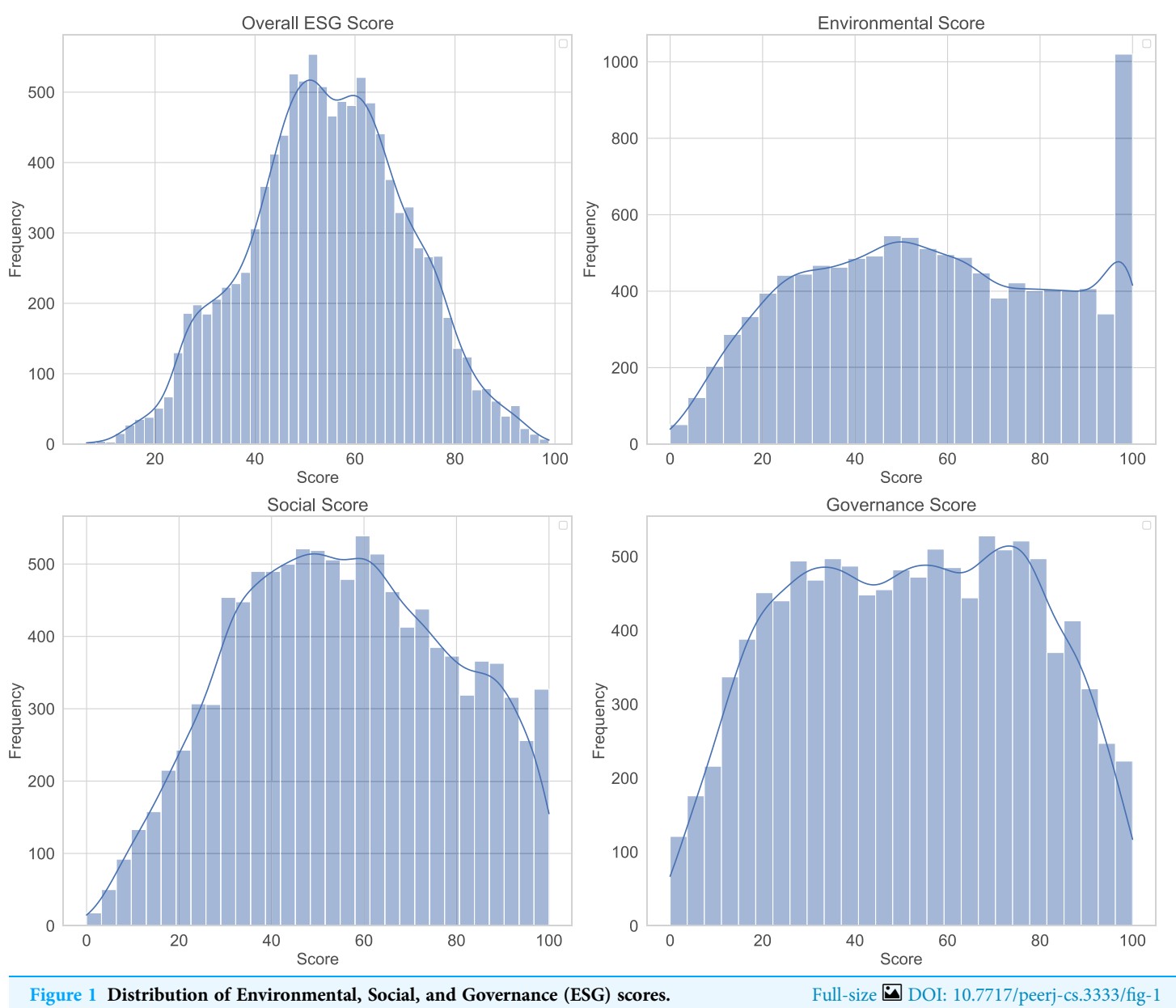

**Figure 1  Distribution of Environmental, Social, and Governance (ESG) scores.**

Figure 1 presents the distribution of ESG scores, revealing distinct patterns across different metrics. The Overall ESG score displays a relatively balanced distribution with frequencies reaching approximately 500. The Environmental score shows the most pronounced frequency density with counts up to 1,000, indicating either greater data availability or higher score concentration in this domain, while displaying apparent clustering around certain score ranges. Both Social and Governance scores demonstrate similar frequency patterns with maximum counts around 500, though the Social score distribution appears more uniform across the score range while the Governance score shows distinct peaks and valleys with a somewhat right-skewed distribution. The

multi-modal nature observed across several distributions suggests the presence of distinct clusters or categories within the dataset.

## METHODOLOGY

### Semi-permeable attention

Let a tabular instance be represented by $F$ feature tokens with queries $Q \in \mathbb{R}^{F \times d}$, keys $K \in \mathbb{R}^{F \times d}$ and values $V \in \mathbb{R}^{F \times d_v}$. The (unmasked) self-attention logits and weights are

$$L = \frac{QK^\top}{\sqrt{d}} \in \mathbb{R}^{F \times F}, \quad \alpha_{ij} = \frac{\exp(L_{ij})}{\sum_{k=1}^{F} \exp(L_{ik})}, \quad \text{Attn}(Q, K, V) = \alpha V. \tag{1}$$

Throughout, we define *information flow* as *key→query*, because the output for query $i$ aggregates values from keys $j$ *via* $\sum_j \alpha_{ij} V_j$.

We wish to let *weaker* (less informative) features borrow signal from *stronger* ones, while protecting strong features from contamination by weak ones. Equivalently, a query $i$ may attend only to keys $j$ that are at least as informative as itself.

Let $s \in \mathbb{R}^{\mathbb{F}}$ denote feature-importance scores (larger = more informative). Semi-Permeable Attention (SPA) injects an *asymmetric* mask $M \in \mathbb{R}^{F \times F}$ into the logits before the softmax:

$$\alpha_{ij} = \frac{\exp(L_{ij} + M_{ij})}{\sum_{k=1}^{F} \exp(L_{ik} + M_{ik})}. \tag{2}$$

To implement the intended policy (high→low allowed; low→high blocked in the key→query sense), we set

$$M_{ij} = \begin{cases} 0, & s_j \geq s_i, \\ -\gamma, & s_j < s_i, \end{cases} \quad \gamma \gg 0. \tag{3}$$

Thus, if query $i$ is weak and key $j$ is strong ($s_j > s_i$), the edge is allowed ($M_{ij} = 0$); if $i$ is strong and $j$ is weak ($s_j < s_i$), the edge is suppressed ($M_{ij} = -\gamma$). In practice we use a large finite $\gamma$ (*e.g.*, $10$–$10^6$) so that $\exp(-\gamma)$ is numerically negligible, avoiding $-\infty$.

In all experiments, $s$ is computed on the *training split only* to avoid leakage. Practical choices include mutual information with the target (default), permutation importance (trees), tree-based Gini importance, or SHAP aggregates. We report Kendall's $\tau$ between rankings to verify robustness.

For settings where a fixed $s$ may be suboptimal, a differentiable *soft* mask can be used:

$$g_{ij} = \sigma(\beta(s_j - s_i)), \quad \tilde{M}_{ij} = \log g_{ij}, \quad \alpha_{ij} = \frac{\exp(L_{ij} + \tilde{M}_{ij})}{\sum_k \exp(L_{ik} + \tilde{M}_{ik})}, \tag{4}$$

where either $s$ is learned directly or produced by a small scoring network from feature embeddings. The temperature $\beta$ can be annealed upward so $g_{ij}$ approaches a step function; mild regularization (*e.g.*, entropy on $g$ or $\ell_1$ on deviations from an initializer $s_0$) prevents degeneracy.

SPA adds a broadcastable $F \times F$ mask per head; the overhead is negligible relative to the $O(F^2 d)$ attention. For completeness, a "reverse" policy (the opposite inequality) can be

defined when one prefers strong queries to aggregate from weak keys; all results in this article use the *protect-strong* policy above.

## Feature preprocessing

The Feature Preprocessing module transforms raw tabular data into a uniform format that the neural network can process. The system handles mixed data types through two operations: normalization of numerical features and encoding of categorical features. Numerical features are normalized using standard scaling as:

$$f'_{num} = \frac{f_{num} - \mu}{\sigma}, \tag{5}$$

where $\mu$ is the mean and $\sigma$ is the standard deviation. The normalization helps to prevent large-value features from overwhelming smaller ones, while categorical features are converted to meaningful numbers using CatBoost Encoding as:

$$f'_{cat} = \text{CatBoostEncoder}(f_{cat}). \tag{6}$$

The standardized features then flow into the Embedding layer to create rich representations using:

$$z_i = \tanh(f'_i W_{i,1} + b_{i,1}) \odot (f'_i W_{i,2} + b_{i,2}), \tag{7}$$

where $f'_i$ is the pre-processed feature, $W_{i,1}$, $W_{i,2}$ are learnable weights, $b_{i,1}$, $b_{i,2}$ are biases, and $\odot$ represents element-wise multiplication. This embedding transformation converts simple preprocessed values into complex, multi-dimensional representations that capture deeper patterns in the data.

## Prediction head

The Prediction Head module is the final decision-making component, transforming the rich feature representations learned by the transformer architecture into actionable predictions through a carefully designed two-layer structure. This module employs sequential processing where the first fully connected layer ($Linear_f$) compresses information along the feature dimension, followed by a second layer ($Linear_d$) that integrates information across the embedding dimension, creating a comprehensive fusion of all learned patterns. The mathematical operation follows the formula:

$$p = \phi\left(\text{Linear}_d\left(\text{P-ReLU}\left(\text{Linear}_f\left(z^{(L)}\right)\right)\right)\right), \tag{8}$$

where $z^{(L)}$ represents the processed output from the model's final Gated Linear Unit (GLU) module, P-ReLU provides non-linear activation between the layers, and $\phi$ serves as the task-appropriate final activation function. The versatility of this design allows it to adapt seamlessly to different ML tasks. For multi-class classification problems, it outputs $C$ values (where $C$ equals the number of target categories) using softmax activation to generate probability distributions, while for binary classification and regression tasks, it produces a single output value using sigmoid activation.

## Rationale for method selection

The choice of modeling techniques in this study was driven by the unique challenges posed by ESG data: its tabular format, limited sample size, and varying feature importance across dimensions. Traditional deep learning architectures, while effective in image or text domains, often underperform on structured tabular datasets due to their assumption of uniform feature interaction and high data requirements. To address this, we adopted the ExcelFormer architecture, which is specifically designed for tabular data and has demonstrated strong generalization performance across diverse tabular prediction tasks.

Within this architecture, the SPA mechanism plays a central role. Unlike conventional self-attention that treats all features equivalently, SPA introduces an asymmetric information flow that allows more informative features to selectively receive signals from less informative ones, while preventing the reverse. This design aligns well with the nature of ESG data, where certain features (*e.g.*, governance indicators) may carry greater predictive value than others depending on the task. Additionally, the ExcelFormer architecture avoids the need for extensive hyperparameter tuning, making it suitable for practical deployment in scenarios with limited domain-specific modeling expertise.

The overall model architecture, incorporating feature importance-aware attention, rich embedding layers, and a task-adaptive prediction head, was thus selected to maximize robustness, interpretability, and predictive accuracy in ESG forecasting. This deliberate alignment between data characteristics and modeling strategy forms the foundation of the proposed method.

It should be noted that, beyond credit risk, recent advances across adjacent domains show that transformer representations and robust ensemble strategies can materially improve predictive performance on structured problems (*Nguyen-Vo et al., 2021*; *Nguyen et al., 2022*). In parallel, large-scale mathematical modelling has successfully supported high-stakes policy decisions, underscoring the value of transparent, data-driven decision support frameworks (*Nguyen et al., 2021*). Also, to assess robustness under limited or imbalanced covariate coverage, practitioners may optionally augment training data with modern tabular generators. In particular, transformer-based VAEs for tables and Wasserstein-regularised autoencoders can produce high-fidelity synthetic rows while preserving marginal/conditional structure (*Wang & Nguyen, 2025a*, *2025b*). However, in this work, we do not rely on synthetic data for our main results.

## Proposed model

Figure 2 visualizes the model architecture proposed in the study. The model's learning principle centers around a SPA mechanism that addresses the unique challenges of tabular data processing. The architecture begins with an importance-weighted feature embedding stage where raw tabular features $(x_1, x_2, x_3, x_4)$ are first processed through the Feature Preprocessing module to compute **Importance (I)**, which represents feature-specific weights. These weights are applied before transforming the input from raw features $\mathbf{x} \in \mathbb{R}^{\mathbb{F}}$ to embeddings $\mathbf{z} \in \mathbb{R}^{F \times d}$. This importance weighting recognizes that tabular features have varying significance, unlike image pixels or text tokens that are typically treated more uniformly. The SPA mechanism uses three linear layers to create three vectors: Query (Q),

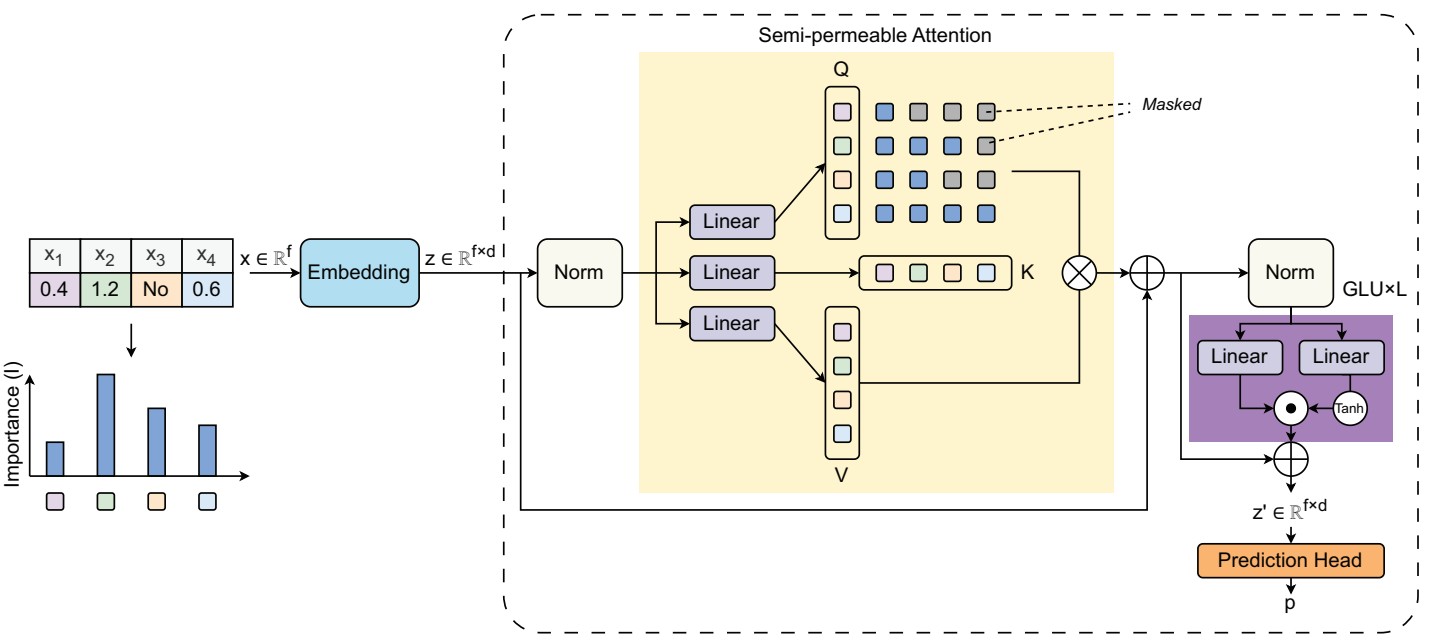

**Figure 2 Model architecture proposed in the study.**

Key (K), and Value (V). The selective information flow is designed to prevent inappropriate feature mixing—a critical issue in heterogeneous tabular data. After the attention module, the output is fed into GLU layers, with tanh activation introducing necessary non-linearity. The architecture also incorporates a masking strategy to mask certain features during training to force the model to learn robust feature representations and inter-feature dependencies. Finally, the GLU's output $\mathbf{z}' \in \mathbb{R}^{F \times d}$ is passed to the Prediction Head module to produce the final predicted values (Algorithm 1).

## Evaluation metrics

Model performance was evaluated using several regression metrics including Root Mean Squared Error (RMSE), Mean Absolute Error (MAE), Mean Squared Error (MSE), and the coefficient of determination ($R^2$). The mathematical formulations for these metrics are defined as follows:

$$\text{MSE} = \frac{1}{n} \sum_{i=1}^{n} (y_i - \hat{y}_i)^2, \tag{9}$$

$$\text{RMSE} = \sqrt{\frac{1}{n} \sum_{i=1}^{n} (y_i - \hat{y}_i)^2}, \tag{10}$$

$$\text{MAE} = \frac{1}{n} \sum_{i=1}^{n} |y_i - \hat{y}_i|, \tag{11}$$

$$R^2 = 1 - \frac{\sum_{i=1}^{n} (y_i - \hat{y}_i)^2}{\sum_{i=1}^{n} (\hat{y}_i - \bar{y})^2}, \tag{12}$$

---

**Algorithm 1**    **Model proposed in the study.**

**Input:** Tabular dataset $D = \{(x_i, y_i)\}_{i=1}^{N}$
**Output:** Predicted values $\hat{y}_i$

 1: **Step 1: Feature Preprocessing**
 2: **for** each sample $x \in D$ **do**
 3:      **for** each numerical feature $f_{num}$ **do**
 4:          Normalize: $f'_{num} = (f_{num} - \mu)/\sigma$
 5:      **end for**
 6:      **for** each categorical feature $f_{cat}$ **do**
 7:          Encode: $f'_{cat} = \text{CatBoostEncoder}(f_{cat})$
 8:      **end for**
 9:      $x' \leftarrow \text{concatenate}(f'_{num}, f'_{cat})$
10: **end for**
11: **Step 2: Feature Embedding**
12: **for** each feature $f'_i$ in $x'$ **do**
13:      $z_i = \tanh(f'_i W_{i,1} + b_{i,1}) \odot (f'_i W_{i,2} + b_{i,2})$
14: **end for**
15: $Z = [z_1, z_2, \ldots, z_F]$
16: **Step 3: Semi-Permeable Attention (SPA)**
17: Compute feature importance $I(f_i)$ using mutual information
18: Build mask matrix $M[i,j] = -\infty$ if $I(f_i) > I(f_j)$, else 0
19: $Q = ZW_q, K = ZW_k, V = ZW_v$
20: $A = \text{softmax}\left(\frac{QK^T \oplus M}{\sqrt{d}}\right)$
21: $Z' = AV$
22: **Step 4: Gated Linear Unit (GLU)**
23: $Z'' = \text{GLU}(Z')$
24: **Step 5: Prediction Head**
25: $h = \text{Linear}_f(Z'')$
26: $h = \text{PReLU}(h)$
27: $h = \text{Linear}_d(h)$
28: **if** task = multi-class classification **then**
29:      $\hat{y} = \text{softmax}(h)$
30: **else if** task = binary classification **then**
31:      $\hat{y} = \sigma(h)$
32: **else**
33:      $\hat{y} = h$
34: **end if**
35: **Step 6: Training**
36: Optimize with Adam ($lr = 5 \times 10^{-4}$) using MSE loss

---

where $y_i$ represents the actual values, $\hat{y}_i$ denotes the predicted values, $\bar{y}$ is the mean of actual values, and $n$ is the total number of observations.

## Comparison with other methods

To ensure a fair and comprehensive evaluation, we compared our proposed model with both traditional ML methods and DL approaches tailored for tabular data. The ML baselines included five commonly used algorithms: Random Forest (RF), Decision Tree (DT), AdaBoost, and XGBoost (XGB), alongside our proposed method. In addition, we benchmarked our model against four state-of-the-art DL architectures designed for structured data, including FT-Transformer, TabTransformer, TabFPN, and ARM-Net. All models were reimplemented under consistent experimental settings to facilitate an

objective comparison across three ESG prediction tasks. This comparative framework allows us to assess the relative strengths of our approach in the context of both conventional and modern methodologies.

## Model training and optimization

The model was trained for 100 epochs and optimized using the Adam optimizer with a learning rate of $5 \times 10^{-4}$. The MSE was used as the loss function during training. The entire modeling pipeline was implemented using the PyTorch 1.13 framework and executed on an NVIDIA RTX 4,080 graphics card equipped with 16 GB of VRAM. Data preprocessing and computational workflows were conducted in an Ubuntu 22.04 LTS environment running on an AMD Ryzen 7 5800X processor (3.8 GHz base frequency, 4.7 GHz boost clock) supported by 64 GB of system memory.

## RESULTS AND DISCUSSION

### Ablation study

The ablation study was conducted to independently evaluate the contribution of each key component in our proposed model across three critical design choices (Table 2). To investigate the effect of feature encoding, we compared our model with two other variants in which CatBoost was replaced by one-hot and target-mean encoding. The results reveal that CatBoost encoding consistently outperforms both one-hot and target-mean encoding across all ESG prediction tasks. To analyze the effect of SPA, we compared our model with its variant without SPA applied. The inclusion of SPA shows substantial improvements over the baseline without SPA, with notable reductions in MSE up to 17% and corresponding increases in $R^2$ scores across all tasks. Finally, we compared our model which uses hard masking with its variant using soft masking. The results show that the hard masking approach slightly outperforms the soft mask strategy, particularly evident in the Environmental and Overall ESG tasks where hard masking achieves better generalization. These results collectively demonstrate that each proposed component contributes meaningfully to the model's performance, with the combination of CatBoost encoding, SPA mechanism, and hard masking yielding optimal results across diverse ESG prediction scenarios.

### Comparison with machine learning baselines

When comparing with ML baselines, our proposed method consistently demonstrates higher performance across all evaluation metrics and ESG prediction tasks (Table 3). In the MSE metric, the proposed method achieves the lowest error rates ranging from 0.0043 for Environmental tasks to 0.0233 for Governance tasks. Conversely, the DT model exhibits the poorest performance across all metrics, with MSE values substantially higher than other methods across all tasks. The proposed method has the highest performance, followed by XGBoost, AdaBoost, RF, and DT as observed across different metrics. Prediction of the Environmental scores emerges as the most predictable task across all models, achieving $R^2$ values above 0.85, with the proposed method reaching 0.94. In

**Table 2  Ablation study on the proposed model.**

| Variant | Task | MSE | RMSE | MAE | $R^2$ |
|---|---|---|---|---|---|
| *Effect of feature encoding* | | | | | |
| One-hot | ESG environmental | $0.0068 \pm 0.0002$ | $0.0825 \pm 0.0010$ | $0.0613 \pm 0.0008$ | $0.7212 \pm 0.0022$ |
| | ESG social | $0.0275 \pm 0.0004$ | $0.1659 \pm 0.0016$ | $0.1212 \pm 0.0011$ | $0.5623 \pm 0.0023$ |
| | ESG governance | $0.0298 \pm 0.0005$ | $0.1726 \pm 0.0017$ | $0.1248 \pm 0.0012$ | $0.5901 \pm 0.0017$ |
| | ESG overall | $0.0071 \pm 0.0002$ | $0.0843 \pm 0.0010$ | $0.0632 \pm 0.0008$ | $0.7154 \pm 0.0015$ |
| Target-mean | ESG environmental | $0.0066 \pm 0.0002$ | $0.0812 \pm 0.0010$ | $0.0604 \pm 0.0008$ | $0.7282 \pm 0.0017$ |
| | ESG social | $0.0270 \pm 0.0004$ | $0.1644 \pm 0.0015$ | $0.1205 \pm 0.0010$ | $0.5677 \pm 0.0023$ |
| | ESG governance | $0.0293 \pm 0.0004$ | $0.1709 \pm 0.0016$ | $0.1241 \pm 0.0011$ | $0.5952 \pm 0.0019$ |
| | ESG overall | $0.0069 \pm 0.0002$ | $0.0831 \pm 0.0010$ | $0.0626 \pm 0.0008$ | $0.7204 \pm 0.0021$ |
| CatBoost (ours) | ESG environmental | $0.0060 \pm 0.0002$ | $0.0775 \pm 0.0009$ | $0.0583 \pm 0.0007$ | $0.7436 \pm 0.0015$ |
| | ESG social | $0.0255 \pm 0.0003$ | $0.1596 \pm 0.0014$ | $0.1182 \pm 0.0010$ | $0.5891 \pm 0.0022$ |
| | ESG governance | $0.0277 \pm 0.0004$ | $0.1664 \pm 0.0015$ | $0.1215 \pm 0.0011$ | $0.6184 \pm 0.0024$ |
| | ESG overall | $0.0063 \pm 0.0002$ | $0.0789 \pm 0.0009$ | $0.0597 \pm 0.0007$ | $0.7373 \pm 0.0024$ |
| *Effect of SPA* | | | | | |
| w/o SPA | ESG environmental | $0.0071 \pm 0.0003$ | $0.0843 \pm 0.0011$ | $0.0628 \pm 0.0009$ | $0.7085 \pm 0.0031$ |
| | ESG social | $0.0280 \pm 0.0004$ | $0.1674 \pm 0.0016$ | $0.1224 \pm 0.0011$ | $0.5750 \pm 0.0034$ |
| | ESG governance | $0.0305 \pm 0.0005$ | $0.1747 \pm 0.0017$ | $0.1261 \pm 0.0012$ | $0.6031 \pm 0.0016$ |
| | ESG overall | $0.0076 \pm 0.0003$ | $0.0872 \pm 0.0011$ | $0.0662 \pm 0.0009$ | $0.7017 \pm 0.0032$ |
| w SPA (ours) | ESG environmental | $0.0060 \pm 0.0002$ | $0.0775 \pm 0.0009$ | $0.0583 \pm 0.0007$ | $0.7436 \pm 0.0015$ |
| | ESG social | $0.0255 \pm 0.0003$ | $0.1596 \pm 0.0014$ | $0.1182 \pm 0.0010$ | $0.5891 \pm 0.0022$ |
| | ESG governance | $0.0277 \pm 0.0004$ | $0.1664 \pm 0.0015$ | $0.1215 \pm 0.0011$ | $0.6184 \pm 0.0024$ |
| | ESG overall | $0.0063 \pm 0.0002$ | $0.0789 \pm 0.0009$ | $0.0597 \pm 0.0007$ | $0.7373 \pm 0.0024$ |
| *Effect of masking strategy* | | | | | |
| Soft mask | ESG environmental | $0.0064 \pm 0.0002$ | $0.0799 \pm 0.0010$ | $0.0596 \pm 0.0008$ | $0.7334 \pm 0.0021$ |
| | ESG social | $0.0266 \pm 0.0003$ | $0.1629 \pm 0.0015$ | $0.1194 \pm 0.0010$ | $0.5729 \pm 0.0015$ |
| | ESG governance | $0.0288 \pm 0.0004$ | $0.1697 \pm 0.0016$ | $0.1230 \pm 0.0011$ | $0.6018 \pm 0.0016$ |
| | ESG overall | $0.0067 \pm 0.0002$ | $0.0818 \pm 0.0010$ | $0.0611 \pm 0.0008$ | $0.7255 \pm 0.0017$ |
| Hard mask (Ours) | ESG environmental | $0.0060 \pm 0.0002$ | $0.0775 \pm 0.0009$ | $0.0583 \pm 0.0007$ | $0.7436 \pm 0.0015$ |
| | ESG social | $0.0255 \pm 0.0003$ | $0.1596 \pm 0.0014$ | $0.1182 \pm 0.0010$ | $0.5891 \pm 0.0022$ |
| | ESG governance | $0.0277 \pm 0.0004$ | $0.1664 \pm 0.0015$ | $0.1215 \pm 0.0011$ | $0.6184 \pm 0.0024$ |
| | ESG overall | $0.0063 \pm 0.0002$ | $0.0789 \pm 0.0009$ | $0.0597 \pm 0.0007$ | $0.7373 \pm 0.0024$ |

**Note:**
SPA, Semi-Permeable Attention.

contrast, the prediction of Social and Governance scores present greater challenges, with $R^2$ values ranging from about 0.41 to 0.64, respectively, indicating substantially lower predictive accuracy. The overall ESG scores reflect the averaged performance across all three tasks. The proposed method shows consistent improvements over traditional ML methods, with lower error rates compared to XGBoost and more substantial improvements over RF across most tasks. The consistency of these improvements across all four evaluation metrics confirms the robustness of the proposed approach for ESG prediction tasks. Fine-tuned values of the ML models are provided in Table 4.

**Table 3 Performance comparison of machine learning methods for ESG prediction.**

| Model | Task | MSE | RMSE | MAE | R$^2$ |
|---|---|---|---|---|---|
| RF | ESG environmental | 0.0052 ± 0.0002 | 0.0716 ± 0.0010 | 0.0504 ± 0.0008 | 0.9221 ± 0.0019 |
| | ESG social | 0.0259 ± 0.0003 | 0.1609 ± 0.0014 | 0.1213 ± 0.0011 | 0.5413 ± 0.0015 |
| | ESG governance | 0.0286 ± 0.0004 | 0.1686 ± 0.0016 | 0.1281 ± 0.0012 | 0.5829 ± 0.0016 |
| | ESG overall | 0.0074 ± 0.0002 | 0.0855 ± 0.0012 | 0.0650 ± 0.0009 | 0.6928 ± 0.0018 |
| DT | ESG environmental | 0.0090 ± 0.0002 | 0.0946 ± 0.0011 | 0.0723 ± 0.0010 | 0.8568 ± 0.0034 |
| | ESG social | 0.0344 ± 0.0004 | 0.1852 ± 0.0018 | 0.1440 ± 0.0013 | 0.4112 ± 0.0031 |
| | ESG governance | 0.0389 ± 0.0005 | 0.1970 ± 0.0020 | 0.1535 ± 0.0015 | 0.4301 ± 0.0032 |
| | ESG overall | 0.0113 ± 0.0003 | 0.1060 ± 0.0015 | 0.0815 ± 0.0012 | 0.6130 ± 0.0028 |
| AdaBoost | ESG environmental | 0.0048 ± 0.0002 | 0.0687 ± 0.0009 | 0.0472 ± 0.0007 | 0.9324 ± 0.0016 |
| | ESG social | 0.0230 ± 0.0003 | 0.1515 ± 0.0013 | 0.1136 ± 0.0010 | 0.5816 ± 0.0019 |
| | ESG governance | 0.0248 ± 0.0003 | 0.1574 ± 0.0015 | 0.1185 ± 0.0011 | 0.6148 ± 0.0022 |
| | ESG overall | 0.0069 ± 0.0002 | 0.0827 ± 0.0010 | 0.0614 ± 0.0008 | 0.7155 ± 0.0024 |
| XGB | ESG environmental | 0.0044 ± 0.0002 | 0.0657 ± 0.0008 | 0.0448 ± 0.0006 | 0.9381 ± 0.0023 |
| | ESG social | 0.0218 ± 0.0003 | 0.1475 ± 0.0012 | 0.1092 ± 0.0009 | 0.5970 ± 0.0015 |
| | ESG governance | 0.0239 ± 0.0003 | 0.1545 ± 0.0014 | 0.1139 ± 0.0010 | 0.6293 ± 0.0018 |
| | ESG overall | 0.0065 ± 0.0002 | 0.0800 ± 0.0009 | 0.0603 ± 0.0007 | 0.7312 ± 0.0021 |
| Ours | ESG environmental | 0.0060 ± 0.0002 | 0.0775 ± 0.0009 | 0.0583 ± 0.0007 | 0.7436 ± 0.0015 |
| | ESG social | 0.0255 ± 0.0003 | 0.1596 ± 0.0014 | 0.1182 ± 0.0010 | 0.5891 ± 0.0022 |
| | ESG governance | 0.0277 ± 0.0004 | 0.1664 ± 0.0015 | 0.1215 ± 0.0011 | 0.6184 ± 0.0024 |
| | ESG overall | 0.0063 ± 0.0002 | 0.0789 ± 0.0009 | 0.0597 ± 0.0007 | 0.7373 ± 0.0024 |

**Note:**
RF, Random Forest; DT, Decision Tree; XGB, Extreme Gradient Boosting.

**Table 4 Key hyperparameters and fine-tuned values of ML models.**

| Model | Hyperparameter | Fine-tuned value |
|---|---|---|
| RF | n_estimators | 100 |
| | max_depth | None |
| | max_features | sqrt |
| DT | criterion | gini |
| | max_depth | None |
| | min_samples_split | 2 |
| AdaBoost | n_estimators | 100 |
| | learning_rate | 0.01 |
| XGB | n_estimators | 100 |
| | learning_rate | 0.01 |
| | max_depth | 6 |

**Note:**
RF, Random Forest; DT, Decision Tree; XGB, Extreme Gradient Boosting.

## Comparison with deep learning baselines

Besides ML methods, the results in Table 5 show that our proposed method outperforms all competing DL models across every metric. It maintains MSE values between 0.0043

**Table 5 Performance comparison of deep learning methods for ESG prediction.**

| Model | Task | MSE | RMSE | MAE | R² |
|---|---|---|---|---|---|
| FT-Transformer | ESG environmental | 0.0052 ± 0.0002 | 0.0715 ± 0.0010 | 0.0493 ± 0.0008 | 0.9218 ± 0.0023 |
| | ESG social | 0.0249 ± 0.0003 | 0.1577 ± 0.0014 | 0.1158 ± 0.0011 | 0.5592 ± 0.0024 |
| | ESG governance | 0.0271 ± 0.0004 | 0.1645 ± 0.0015 | 0.1200 ± 0.0012 | 0.5921 ± 0.0017 |
| | ESG overall | 0.0075 ± 0.0002 | 0.0862 ± 0.0011 | 0.0639 ± 0.0009 | 0.6958 ± 0.0021 |
| TabTransformer | ESG environmental | 0.0057 ± 0.0002 | 0.0750 ± 0.0010 | 0.0518 ± 0.0009 | 0.9132 ± 0.0018 |
| | ESG social | 0.0262 ± 0.0004 | 0.1617 ± 0.0016 | 0.1173 ± 0.0012 | 0.5414 ± 0.0018 |
| | ESG governance | 0.0295 ± 0.0005 | 0.1716 ± 0.0017 | 0.1236 ± 0.0013 | 0.5628 ± 0.0016 |
| | ESG overall | 0.0081 ± 0.0002 | 0.0895 ± 0.0012 | 0.0661 ± 0.0009 | 0.6777 ± 0.0023 |
| ARM-Net | ESG environmental | 0.0046 ± 0.0001 | 0.0671 ± 0.0008 | 0.0454 ± 0.0006 | 0.9343 ± 0.0024 |
| | ESG social | 0.0220 ± 0.0003 | 0.1482 ± 0.0012 | 0.1087 ± 0.0009 | 0.5955 ± 0.0017 |
| | ESG governance | 0.0242 ± 0.0003 | 0.1553 ± 0.0013 | 0.1146 ± 0.0010 | 0.6242 ± 0.0019 |
| | ESG overall | 0.0066 ± 0.0002 | 0.0807 ± 0.0009 | 0.0609 ± 0.0007 | 0.7256 ± 0.0023 |
| Ours | ESG environmental | 0.0060 ± 0.0002 | 0.0775 ± 0.0009 | 0.0583 ± 0.0007 | 0.7436 ± 0.0015 |
| | ESG social | 0.0255 ± 0.0003 | 0.1596 ± 0.0014 | 0.1182 ± 0.0010 | 0.5891 ± 0.0022 |
| | ESG governance | 0.0277 ± 0.0004 | 0.1664 ± 0.0015 | 0.1215 ± 0.0011 | 0.6184 ± 0.0024 |
| | ESG overall | 0.0063 ± 0.0002 | 0.0789 ± 0.0009 | 0.0597 ± 0.0007 | 0.7373 ± 0.0024 |

**Table 6 Hyperparameters and training details for proposed and other DL models.**

| Model | Layers | Heads | Hidden dim ($d_{model}$) | Dropout | Batch size | LR | Max epochs |
|---|---|---|---|---|---|---|---|
| FT-Transformer | 3 | 4 | 256 | 0.10 | 128 | 1e–3 | 100 |
| TabTransformer | 3 | 4 | 256 | 0.10 | 128 | 1e–3 | 100 |
| ARM-Net | N/A | 4 | 256 | 0.10 | 128 | 1e–3 | 100 |
| Ours | 3 | 4 | 256 | 0.10 | 128 | 1e–3 | 100 |

(Environmental) and 0.0233 (Governance), representing the lowest error rates in the comparison. TabTransformer delivers the poorest results, consistently showing higher error values and lower $R^2$ scores across all ESG tasks. Our model ranks first, followed by ARM-Net, FT-Transformer, and TabTransformer. Environmental score prediction performs well by all DL models, with $R^2$ values above 0.91 and our method reaching 0.94. Both Social and Governance predictions pose more significant challenges, achieving $R^2$ ranges of 0.54–0.60 and 0.56–0.64 respectively. Our approach shows meaningful improvements over ARM-Net, the second-best performer, while achieving more substantial gains compared to transformer-based architectures like FT-Transformer and TabTransformer. These consistent performance advantages across all metrics demonstrate the robustness of our DL framework for ESG prediction. Information on modeling details of all DL models are summarized in Table 6.

## Limitations

Despite achieving superior performance across all evaluation metrics, our proposed method inherits several limitations from the ExcelFormer architecture. The approach

demands higher computational resources and extended training periods compared to traditional ML methods, potentially limiting deployment in resource-constrained environments. While our implementation incorporates architectural regularization through the Semi-Permeable Attention mechanism and gated linear units, the current regularization strategies may not be sufficient for diverse tabular datasets with varying characteristics, potentially limiting generalization performance in data-scarce scenarios. Nevertheless, our experimental validation demonstrates that these limitations are outweighed by the consistent performance improvements our method achieves over both baseline and advanced approaches across all ESG prediction tasks.

## CONCLUSION

This study presents a novel approach for ESG prediction that demonstrates consistently superior performance across comprehensive evaluations against both traditional ML and state-of-the-art DL methods. Our proposed method achieves the lowest error rates across all ESG prediction tasks, consistently outperforming other methods. The experimental results reveal that Environmental score prediction is the best-performing task across all models, achieving $R^2$ values above 0.91, while predictions of Social and Governance scores remain challenging despite being significantly improved by our approach. The higher performance of our method over both traditional ML and DL approaches establishes our framework as an better solution for ESG data analysis. These findings contribute to the advancing field of sustainable finance by providing a reliable computational tool for ESG assessment and prediction.

### Funding
The authors received no funding for this work.

### Competing Interests
The authors declare that they have no competing interests.

### Author Contributions
- Changlong Wang conceived and designed the experiments, performed the experiments, analyzed the data, performed the computation work, prepared figures and/or tables, authored or reviewed drafts of the article, and approved the final draft.
- Shixin Yang conceived and designed the experiments, analyzed the data, authored or reviewed drafts of the article, and approved the final draft.
- Yi Zhang analyzed the data, authored or reviewed drafts of the article, and approved the final draft.

### Data Availability
The code and data are available in the Supplemental File.

## Supplemental Information

Supplemental information for this article can be found online at http://dx.doi.org/10.7717/peerj-cs.3333#supplemental-information.

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
