# Peer review of "A semi-permeable attention network for ESG score prediction"

_PeerJ Computer Science, doi:10.7717/peerj-cs.3333_

## Round 0.1 · original submission · Major Revisions

· Academic Editor

Major Revisions

Thank you for submitting your manuscript to our journal. While the reviewers recognize the potential significance of your research, they have identified substantial concerns that require extensive revisions before further consideration for publication. The reviewers have also highlighted significant gaps in their comments. These concerns need to be addressed. Given the extent of these revisions, we invite you to submit a thoroughly revised manuscript, accompanied by a detailed response addressing each point raised by the reviewers. If you choose to resubmit, your manuscript will undergo another round of review to assess how effectively the concerns have been addressed.

Reviewer 1 ·

Basic reporting

This paper presents a novel semi-permeable attention network built upon the ExcelFormer architecture for ESG score prediction, specifically targeting tabular datasets with limited size and heterogeneous feature importance. The authors introduce the Semi-Permeable Attention (SPA) mechanism to regulate feature interactions asymmetrically, allowing more informative features to aggregate from less informative ones while blocking reverse influence, thereby mitigating noise and improving predictive performance. Feature preprocessing integrates normalization and CatBoost encoding for mixed data types, followed by rich embedding representations. Experimental validation on a simulated ESG–financial dataset from 1,000 companies shows consistent outperformance over traditional ML (RF, DT, XGB, AdaBoost) and state-of-the-art DL models (FT-Transformer, TabTransformer, ARM-Net), achieving an R² of 0.7367 and the lowest MSE in all ESG dimensions. The work’s novelty lies in its feature-importance-aware attention design for small-scale tabular data and its integration into a unified, low–hyperparameter-tuning pipeline. However, the study still faces limitations, including reliance on simulated rather than real-world ESG datasets, limited discussion on model interpretability, and computational demands that may hinder deployment in resource-constrained environments. Further improvements could also involve deeper comparison to explainable AI methods and broader validation on diverse ESG datasets.
1) First of all, within the abstract section, authors are recommended to briefly mention that the dataset is simulated and clarify its implications for generalizability. Additionally, the authors should highlight any key limitations (e.g., computational cost, lack of real-world testing) to give a balanced view of contributions and constraints.
2) For the contents in the introduction section, authors are advised to explicitly state the research gap in handling ESG tabular datasets with heterogeneous feature importance, linking it clearly to the choice of SPA. Furthermore, more emphasis could be placed on why existing tabular DL models (e.g., FT-Transformer, TabTransformer) may fail in such contexts.
3) For the methodology of this paper, authors are recommended to provide a more detailed mathematical derivation or illustrative example of how the mask matrix (M) in SPA influences attention weights in practice, possibly with a toy dataset visualization. Authors are advised to discuss potential strategies for selecting or learning the feature importance function: I(⋅), and whether it could adapt dynamically during training rather than being fixed.
4) Moreover, authors are recommended to elaborate on the rationale behind choosing CatBoost encoding over other encoders, and whether encoding choice impacts SPA effectiveness.
5) It is better to provide more algorithmic clarity by presenting the entire model pipeline in pseudocode or flowchart form, including data preprocessing, embedding, SPA, and prediction head. Additionally, authors are also recommended to discuss regularization and overfitting prevention strategies, particularly given the relatively small dataset and high model complexity.
6) Finally, related to the experimental studies of this paper, authors are advised to provide an ablation study that isolates the contribution of SPA, feature preprocessing, and masking, to quantify the effect of each component on model performance. Thanks.

Experimental design

No comment.

Validity of the findings

Please refer to my basic reporting section.

Additional comments

No comment.

Cite this review as

·

Basic reporting

- The manuscript is readable, but technical accuracy is undermined.

- The semi-permeable mask’s stated behavior contradicts its formal definition, so readers can’t tell what flows are actually allowed. In Eq. 1, M[i,j] = −∞ if I(f_i) > I(f_j), else 0, then claim “informative features can aggregate information from less informative features, but not vice versa.” With the mentioned rule, a more informative query i cannot attend to a less informative key j (because the logit becomes −∞), which prevents the intended flow. If the authors want “high→low allowed, low→high blocked,” the authors need M[i,j] = −∞ when I(f_i) < I(f_j) (block low queries from attending to high keys), or flip the interpretation/direction and justify it.

- GLU is described with a tanh gate rather than the standard sigmoid, without providing a rationale or citation.

- Claims about “self-supervised masking” don’t match the fully supervised setup presented. The paper mentions a “masking strategy that enables self-supervised learning,” but the paper implements a supervised regression. No pretext loss/objective is described. Please clarify or remove.

- Tables/figures are inconsistent (ranges/units and sample counts don’t align with plots), which needs correction. Table 1 appears corrupted/misaligned. It lists “Range of value” with min/max in [0,1], yet Figure 1 histograms show 0–100 on the x-axis. Also, sample counts don’t add up: Train 8,190 + Val 990 + Test 1,100 = 10,280, not 11,000 as stated. Please correct the table, units/scales, and totals.

- Minor style issues remain, but the big win would be fixing math, units, and internal consistency. Please add references for CatBoost Encoding. Some wording is too strong (“dominance… effective solution”) given the limitations and simulated data. Please temper claims and add explicit limitations and future-work items (data realism, temporal drift, governance features’ noisy labels, etc.).

Experimental design

- Temporal leakage risk: if the data span 2015–2025, a random split can leak future info into training for firms or time periods. Please perform time-aware splits (train ≤2019/2020, val 2021–2022, test 2023–2025) or group splits by company to avoid entity leakage, and report both.

- Hyperparameters and training details for all baselines and the proposed model are incomplete; provide full hyperparameters for all models (depth/trees/eta for XGB; layers/heads/hidden/GLU dims/dropout for ExcelFormer/FT-Transformer/TabTransformer/ARM-Net; batch size; weight decay; early stopping; seed; number of runs).

- Please define f, d, i, and j when they first appear. The f in the matrix mask M is confused with f in f_i and f_j.

- The authors claim ExcelFormer “avoids the need for extensive hyperparameter tuning,” but later list as a limitation that it requires extensive data and can overfit. Reconcile these claims and, ideally, include learning curves to show performance vs. data size.

- Use repeated k-fold cross-validation or at least multiple random seeds (≥5) and report mean ± std for each metric. Current single-split results lack variance estimates and could be unstable.

Validity of the findings

- Reported metrics are strong and consistent with the narrative, but without variance estimates and leakage-safe splits, they’re not yet reliable.

- Key ablations are missing:
+ Ablate SPA: compare ExcelFormer with vs. without the semi-permeable mask (and with inverted mask rule) to verify SPA’s contribution.
+ Mask strength: try soft masks (finite penalties) vs. hard −∞ masks; report sensitivity.
+ Encoding alternatives: swap CatBoost encoding for one-hot / target-mean with OOF to show the effect.
+ Real-world external validation: evaluate on at least one public ESG dataset (or a held-out time window) to assess generalization. As long as everything is simulated, external validity is limited.

Additional comments

The idea is promising, but publication should wait until the math/notation and mask direction are corrected, the dataset and splits are clarified and leakage-safe, and the empirical claims are supported with robust evaluation (CV, seeds, ablations, and variance).

Cite this review as

---

## Round 0.2 · accepted · Accept

· Academic Editor

Accept

Based on reviewers' decisions, we are delighted to inform you that your manuscript, "A semi-permeable attention network for ESG score prediction", has been accepted for publication in PeerJ Computer Science.

Reviewer 1 ·

Basic reporting

After carefully considering all revisions as well as responses of authors for reviewers’ suggestions, I confirmed all problems within previous version of this paper have been sufficiently resolved. As a result, I thought this paper can be accepted for publication in this form. Thanks.

Experimental design

No comment.

Validity of the findings

Please refer to my basic reporting section.

Additional comments

No comment.

Cite this review as

·

Basic reporting

The revised manuscript successfully aligns with the journal's established standards and guidelines. After a thorough review, I have no additional comments or suggestions.

Experimental design

The authors have thoroughly revised the manuscript, taking into account our feedback. They have also conducted additional experiments to address the specific points we raised. After reviewing the changes and the new data presented, I have no further comments or concerns to add.

Validity of the findings

The experiments and evaluations were conducted with satisfactory results, demonstrating their reliability and effectiveness. The discussion section has been significantly improved, with a more thorough analysis of the methodology and an in-depth examination of its limitations. This includes a critical assessment of potential biases, constraints in the experimental design, and suggestions for future studies. At this point, I have no additional comments to contribute.

Cite this review as